

# Validity of using perceived exertion to assess muscle fatigue during resistance exercises

Hanye Zhao[1], Takuya Nishioka[1] and Junichi Okada[2]

[1] Graduate School of Sport Sciences, Waseda University, Tokorozawa, Saitama, Japan
[2] Faculty of Sport Sciences, Waseda University, Tokorozawa, Saitama, Japan

## ABSTRACT

**Background:** The rating of perceived exertion (RPE) is correlated with physiological variables. The purpose of this study was to assess the validity of using the Borg CR-10 scale and velocity to predict muscle fatigue assessed by surface electromyography during single joint resistance exercises.

**Methods:** Fifteen healthy males underwent different fatigue levels of unilateral elbow flexion (EF) and knee extension (KE), consisting of low, medium, and high volumes at 65% of their one-repetition maximum. The RPEs, spectral fatigue index (SFI), and mean velocity of the experimental exercises were assessed throughout the trials.

**Results:** Significant differences in overall RPE ($p < 0.001$) and average SFI ($p < 0.001$) were observed between the conditions in both exercises. Significant changes in RPE and SFI ($p < 0.001$) were observed throughout the EF, whereas a SFI increase ($p < 0.001$) was only observed at the end point of KE. Multiple regression analyses revealed two significant models ($p < 0.001$) for the prediction of muscle fatigue during EF ($R^2 = 0.552$) and KE ($R^2 = 0.377$).

**Conclusions:** Muscle fatigue resulted in similar increases in perceptual responses, demonstrating that RPE is useful for assessing fatigue when resistance exercise is performed. However, velocity changes may not reflect muscle fatigue correctly when exercise is no longer performed in an explosive manner. We recommend combining RPE responses with velocity changes to comprehensively assess muscle fatigue during clinical and sports situations.

Corresponding author
Hanye Zhao,
zhaohanye@toki.waseda.jp

## INTRODUCTION

Fatigue appears in daily life and can be defined as a sensation of tiredness and weakness, underpinned by many physiological and psychological processes (*González-Izal et al., 2012*; *Azevedo et al., 2021*). In sports and rehabilitation situations, exercise-induced fatigue is an inevitable factor. It is considered to be related to sports performance because of the accompanying impairments in the force and/or power generating capacity (*Sparto et al., 1997*). Muscle fatigue during exercise can also increase injury risks (*Borotikar et al., 2008*). Muscle fatigue could be quantified by many physiologically relevant measures, such as blood lactate and muscle force (*Skurvydas et al., 2010*; *Sánchez-Medina &*

*González-Badillo, 2011*). Surface electromyography (sEMG) is a method for non-invasively assessing neuromuscular fatigue responses and is widely used in sports science (*Hermens et al., 2000*; *González-Izal et al., 2012*; *Campanini et al., 2020*). Neuromuscular changes that occur due to fatigue can be measured by analyzing signals collected during muscle contractions (*González-Izal et al., 2010*; *Campanini et al., 2020*). sEMG signals have been shown to provide reliable information regarding the mechanism of muscle fatigue (*Yoshitake et al., 2001*). In using sEMG for muscle fatigue assessment, the manifestation of fatigue is defined as compression of the power spectrum, which is induced by physiologically relevant factors (such as muscle fiber conduction velocity) (*Brody et al., 1991*). The compression of the power spectrum causes a decrease in mean and median frequencies toward lower spectral frequencies, that are generally considered fatigue indicators (*Campanini et al., 2020*).

It is crucial for personal trainers, coaches, and physical therapists to grasp the fatigue conditions of their subjects. However, it is not realistic for them to have their own sEMG devices because these devices are very expensive. Even if sEMG assessment was available, real-time feedback of fatigue is required in many exercise scenarios (such as rehabilitation), which cannot be achieved by monitoring spectral changes as they occur over a much longer time scale (*Dimitrov et al., 2006*). Moreover, sEMG signal spectral characteristics are strongly affected during dynamic contractions, and the measurements of the myoelectric manifestations of muscle fatigue in dynamic contractions require considerable competence and caution (*Dimitrov et al., 2006*; *Campanini et al., 2020*).

Recently, velocity assessment has become very common in resistance exercise situations, and a wide range of tools are currently available for velocity monitoring (*Evandro et al., 2019*; *John et al., 2021*). Based on the easy-to-use characteristics of velocity measures, velocity changes (the transient decline in velocity due to the reduction in force generating capacity) and velocity loss (the percent loss in velocity from the fastest to the slowest repetition of each set) have been recommended as new fatigue indicators for resistance exercises (*Sánchez-Medina & González-Badillo, 2011*; *Mayo, Iglesias-Soler & Kingsley, 2019*). Evidence has been presented that these velocity-based parameters are significantly correlated with metabolic and mechanical measurements of fatigue during resistance exercises (*Sánchez-Medina & González-Badillo, 2011*; *Mayo, Iglesias-Soler & Kingsley, 2019*). However, most sports science studies on velocity changes have only examined the relationship between velocity and muscle fatigue when exercise is performed in an explosive manner. Thus, velocity changes may not reflect muscle fatigue correctly in non-explosive settings. For example, for hypertrophy-aimed training programs, the cadence of repetitions is always controlled because the time the muscle spends under mechanical tension is critically important for muscle protein synthesis (*Vargas-Molina et al., 2020*). Under this setting, velocity changes may be inappropriate for muscle fatigue assessment. Further, for beginners and athletes with injuries, it is dangerous for them to perform explosive exercises because of the lack of strength, proper technique, and joint stability (*Thomas & Roger, 2008*). Consequently, the validity of velocity changes as a muscle fatigue indicator might not be justifiable or adequate when explosive performance is no longer the aim of exercise.

The rating of perceived exertion (RPE) scale is a perceptual-based assessment method that uses a combination of numbers, verbal, and pictorial descriptors (*Borg, 1998*; *Robertson et al., 2003*, *2004*). The RPE has been reported to relate to physiological responses, such as muscle activation and oxygen consumption during aerobic exercise (*Robertson et al., 2004*; *Fontes et al., 2010*). In recent years, the RPE has been widely used in resistance exercise to quantify the intensity of exercises. For example, the RPE is reported to relate to the percentage of one-repetition maximum (1RM) lifts during resistance exercise (*Lagally et al., 2002a*; *Lagally et al., 2002b*; *Eston, James & Evans, 2009*). More recently, task-specific RPE scales have also been developed and can be used for prescriptions of both aerobic and resistance exercises (*Robertson et al., 2003*, *2004*). Despite the differences between these RPE scales, it is thought to be a valid and easy-to-use tool for prescribing resistance exercise (*Gearhart et al., 2001*; *Robertson et al., 2003*). However, the association between RPE and muscle fatigue has only been examined for some isometric exercises because the sEMG signal is thought to be unstable during dynamic exercise situations (*Troiano et al., 2008*; *Otto, Emery & Côté, 2018*; *Cruz-Montecinos et al., 2019*).

Recently developed mathematical simulation-based methods for muscle fatigue assessment using sEMG signals are now available (*Dimitrov et al., 2006*; *González-Izal et al., 2012*). Thus, the relationship between RPE and muscle fatigue induced by dynamic contraction might be evinced by using the new sEMG processing technique. Assessing and predicting exercise-induced muscle fatigue through a simple and effective method would be beneficial, as such a method would allow coaches to monitor the fatigue condition of athletes to avoid both acute and chronic injury risks.

The interdependence between physiological and perceptual responses during exercises indicate that the RPE may be a valid method of assessing muscle fatigue during resistance exercises. Moreover, the correlation between muscle fatigue and RPE might be demonstrated by using well-suited sEMG measurements. Accordingly, this study aimed to examine the validity of using RPE to assess muscle fatigue processed by new sEMG parameters and develop a prediction model for muscle fatigue during single-joint resistance exercise. We hypothesized that RPE would exhibit similar responses corresponding to muscle fatigue levels. Further, muscle fatigue could be predicted using the RPE score and velocity obtained during single-joint resistance exercises.

## MATERIALS AND METHODS

### Participants

The sample size was calculated with a statistical power effect size of 0.4 and alpha of 0.05; power was determined to be 0.95 (*Cruz-Montecinos et al., 2019*). Thus, a minimum of 14 subjects was indicated for this study. Accordingly, 15 healthy male adults with no neuromuscular disorders or skeletal muscle injuries and under no medication were recruited to participate in this investigation. Participants were informed about the experimental protocols, measurement items, potential risks, possible discomfort, and benefits of the study, and then provided written consent to participate in this study. The study was developed in accordance with the ethical guidelines of the Declaration of

Helsinki, and this experiment was approved by the Waseda University Human Ethics Committee (No. 2020-042).

## Experimental procedures

This study used a randomized, crossover, repeated-measures design. The experiment was undertaken in two separate sessions (separated by at least 24 h). During the initial session, the instructions of Borg's CR-10 scale were explained to participants. We then obtained descriptive information on each participant. The 1RM of the experimental exercise was measured. The participants were then familiarized with the experimental protocols. During the experimental session, the anchoring procedure, which determined the range of subjective feelings during experimental conditions, was performed first. Three experimental conditions consisting of 30% (Low, L), 60% (Medium, M), and 90% (High, H) volume were performed in random order. The RPE score, sEMG signal, and velocity were recorded throughout the experimental conditions.

## Orientation and familiarization session

During the orientation session, the purpose, experimental protocols, measurement items, potential risks, possible discomfort, and benefits were explained to the participants. The descriptive characteristics of each participant were measured using a bioelectrical impedance device (InBody 720 body composition analyzer; Biospace Co. Ltd, Seoul, South Korea). The length of the limb was also obtained to calculate the locations of the sEMG electrodes. The unilateral biceps curl and unilateral leg extension exercises were selected as the elbow flexion (EF) and knee extension (KE) exercises. The 1RM of the EF and KE exercises were measured for each arm and leg, and each participant completed the 1RM test of the right arm, left arm, right leg, and left leg separately. The EF exercise was performed using a dumbbell (UESAKA T.E Co., LTD, Tokyo, Japan). The EF was performed with participants in a standing position, feet shoulder width apart. The participants' backs were positioned as flat as possible against the wall, and their upper arms were in contact with the wall throughout the lifting procedure. The concentric action started with the elbows in full extension and the dumbbell at the closest point to the floor. When the concentric phase commenced, participants were requested to flex the elbow until maximum flexion occurred. This was followed by the eccentric phase, and the dumbbell was lowered back toward full extension (180°) to complete a full repetition (*Lagally et al., 2002b*; *Robertson et al., 2003*; *Eston, James & Evans, 2009*). The 1RM of EF was determined from the weight of the dumbbell (to the nearest 1 kg). The KE was performed using a leg extension machine (Prime Fitness, Franklin, PA, USA) (*Robertson et al., 2003*). The back and hip were positioned flat against the back support and the seat, and participants were asked to grip the stabilizing handle near the seat throughout the trial. The seat position and back support were adjusted for each subject to ensure knees were at 90° flexion at the start of the concentric phase. Participants were instructed to extend their knee as far as possible during the concentric phase and to control the descent of the leg during the eccentric phase. The 1RM of KE exercise was determined from the weight stack (to the nearest 5 lbs.). The processes were conducted according to

guidelines set by the National Strength and Conditioning Association (*Thomas & Roger, 2008*). This included the warm-up sets, which were followed by a progressive increase in training load until participants successfully completed their maximum effort of one repetition. After the basic measurement, Borg's CR-10 scale was administered to each participant. Borg's CR-10 scale was used because it is one of the most common RPE scales (*Noble et al., 1983*). This scale ranges from 0 (no exertion at all) to 10 (maximal exertion) and includes standard verbal anchors of perception of effort for intermediate values (*Borg, 1998*). The scale's instructions include the following information: "In the next experimental session, you will be asked to perform a series of upper and lower body resistance exercises. During the exercises, we will use this scale to assess your subjective exertion during exercise. The perception of physical exertion is defined as the subjective intensity of effort, strain, discomfort, and fatigue that you feel during exercise (*Lagally & Robertson, 2006*). This scale comprises verbal anchors and numbers. Numbers from 0 to 10 represent the range of your subjective feelings from no exertion at all to maximal exertion. The verbal descriptors next to the numbers will help you to describe the feeling. The numbers should represent your feelings in the limb you have just lifted. For example, if you are asked to lift with your right arm, consider only the feeling in your right arm when reporting your subjective exertion. When reporting your subjective exertion value, please select the nearest number that corresponds to your feeling." After this instruction, the lifting cadence and RPE reporting practice were demonstrated to all the participants. During the practice, lifting cadence was determined with a cadence of a 2 s eccentric phase, 1 s of concentric phase, and 2 s of pause between repetitions. The cadence was controlled by a metronome, and participants were asked to report their current feelings repetition-by-repetition using numbers from the CR-10 scale during the pause. The scale was placed where it could be readily viewed by the participants.

## Experimental session

Prior to beginning the experimental trial, a series of warm-up exercises were demonstrated to all the participants. The warm-up started with a 5 min cycle ergometer exercise at 50 W within the range of 60 rpm. After the ergometer exercise, warm-up lifts were provided for both limbs. For each arm, participants performed three sets of unilateral bicep curls that comprised six repetitions at 50% of the 1RM. For each leg, three sets of unilateral leg extensions were performed for six repetitions at 50% of the 1RM. Overall, each participant performed six sets of unilateral bicep curls and six sets of unilateral leg extensions. After the warm-up procedure, the experimental trial was performed in random order. The relative intensity for EF and KE exercise was 65% of 1RM (*Robertson et al., 2003*). For both exercises, the lifting cadence was determined with a cadence of a 2 s eccentric phase, 1 s of concentric phase, and 2 s of pause between repetitions controlled by a metronome. As the exercises advanced, muscle fatigue occurred, and the pace of repetition was difficult to maintain. Participants may unconsciously shorten the pause and swing their limbs slightly to catch the cadence of the concentric phase. To avoid the influences caused by these unnoticeable movements, the participants were asked to try their best to follow the metronome during the eccentric phase and to pause between repetitions.

For both exercises, the first set was the anchoring procedure, which determines the range of subjective exertion during later trials. The onset of muscular failure was reported to become the main determinant of perceived exertion (*Hackett et al., 2012*; *Mayo, Iglesias-Soler & Fernández-Del-Olmo, 2014*; *Hiscock et al., 2016*). Accordingly, the physical failure was anchored to an RPE of 10 on the CR-10 scale. Participants were asked to perform a single set of unilateral EF (or unilateral KE) until physical failure. The following instructions were given to the participants: "You will undergo a single set of unilateral bicep curls (or unilateral leg extensions) to establish the range of exertion during later trials. The perception of physical exertion is defined as the subjective intensity of effort, strain, discomfort, and fatigue that you feel during exercise. The perception of exertion when you are sitting down in a relaxed state before any physical activity is equivalent to a score of 0, which means 'nothing at all' (*Eston, James & Evans, 2009*). We will ask you to perform a single set of unilateral bicep curls (or unilateral leg extensions) until physical failure, which means you cannot lift another repetition. When you reach failure, the perceptions of exertion are equivalent to a score of 10, which represents feeling 'extremely strong' (*Mayo, Iglesias-Soler & Kingsley, 2019*). You need to remember this range of feelings, and we will ask you to report your score of perceived exertion during later trials based on the range. You need to maintain the cadence of 1 s of the raising phase, 2 s of the lowering phase, and 2 s of pause between repetitions as closely as possible." This set was performed on random arms and legs, and the volume was recorded and used to calculate the volume of the L, M, and H volume conditions. After the anchoring trial, participants were allowed a 5-min rest before the next trial.

After the anchoring trial, three experimental condition trials were conducted on the arm and leg that did not perform the anchoring trial. For example, participants would perform three experimental trials on the right arm if the anchoring trial were performed on the left arm. When performing the experimental trials, three experimental conditions were executed randomly. Participants were not informed of the required repetitions beforehand; however, they were informed of the penultimate repetition and told to stop immediately after the last repetition. Participants were asked to report their feelings of exertion by using a number from the CR-10 scale repetition-by-repetition during the 2 s pause between repetitions. As the exercises advanced, muscle fatigue occurred, and the pace of repetition was difficult to maintain. Under this condition, the participants were asked to try their best to follow the metronome during the eccentric phase and to pause between repetitions. After each condition, the overall RPE was also obtained from participants, which represented the total perceived exertion of the latest trial. The following instructions were given to participants before the trial: "You will undergo several sets of unilateral bicep curls (or unilateral leg extensions) on different arms (or legs). You will not be told the required repetitions before the trial until you have reached that number. We will inform you at the second to last repetition. After that, you can stop the exercise after you finish the last repetition. During the trial, you need to maintain a cadence of 1 s of the raising phase, 2 s of the lowering phase, and 2 s of pause between repetitions as closely as possible. During the pause, you need to report the exertion score of the latest repetition by using a number from the CR-10 scale. After the trial, we will ask you

to report your overall exertion of the latest trial. This exertion score should be considered based on the range of subjective feelings you establish in failure situations and should be as accurate as possible. When reporting the score, make sure to only consider the exercising limb." A 5-min rest interval was allowed between trials (*George et al., 2018*). Before the beginning of the next trial, participants were asked about their RPE to ensure that the subjective exertion had returned to 0. If not, the participants were allowed a longer rest. Overall, all participants reported a 0 from the CR-10 scale, and a 5-min rest interval was provided to all participants.

## Surface electromyography

The sEMG signals for the biceps brachii (long head), brachioradialis, triceps brachii (lateral head), vastus medialis, vastus lateralis, and biceps femoris muscles were obtained using bipolar surface electrodes (ADMEDEC Co., Ltd, Tokyo, Japan). Two Ag/AgCl electrodes were used for each muscle, and the inter-electrode distance was 1 cm. The skin was prepared for the placement of the surface electrodes by shaving, abrasion with sandpaper, and then cleansing the skin using alcohol swabs (*Hermens et al., 1999*). The location of electrodes was determined based on the recommendations by *Barbero, Merletti & Rainoldi (2012)* to avoid the innervation zone. The signals were recorded using an active differential preamplifier configuration and then transferred to a telemetry device (MARQ MQ-8; Kissei-Com Tech, Nagano, Japan). The sampling frequency was 1,000 Hz. The signals were processed with an analog digital converter, amplified, and transferred to a computer. Further, a camera (FMVU-03MTC-CS; FLIR Systems, Inc., Victoria, Canada) was connected to the computer, and the sEMG signals were synchronized with the motion during exercises. The raw signal was then divided into a single repetition and exported for subsequent analysis.

A fourth-order Butterworth band-pass filter (20–450 Hz) was designed to filter noise (*Otto, Emery & Côté, 2018*). This filtered signal was then used to calculate muscle fatigue. sEMG spectral characteristics are strongly affected during dynamic contraction, as discussed in the Introduction section (*Gerdle, Larsson & Karlsson, 2000*; *Dimitrov et al., 2006*). To address this problem, new, highly sensitive spectral fatigue index (SFI) was adopted to assess the muscle fatigue level (*Dimitrov et al., 2006*) (The authors used the acronym "$FI_{nsm5}$" based on mathematical reasoning. We changed the acronym "$FI_{nsm5}$" to "SFI" to enhance the readability of this paper). SFI provides reliable evaluation of muscle fatigue during dynamic contractions comparable with the traditional sEMG spectral characteristics. A fast Fourier transformation was applied to calculate the power density spectrum. Spectral moments were used to extract the characteristic features of the power spectral density function and were calculated using the following formula:

$$M_k = \int_{f_{min}}^{f_{max}} f^k \cdot PS(f) \cdot df \tag{1}$$

where Mk is a spectral moment of order $k$, PS(f) denotes the power frequency spectrum as a function of frequency $f$, and $f_{min}$ and $f_{max}$ delineate the bandwidth of the signal. SFI was calculated as the ratio between orders −1 and 5, based on the following formular:

$$SFI = \frac{\int_{f_{min}}^{f_{max}} f^{-1} \cdot PS(f) \cdot df}{\int_{f_{min}}^{f_{max}} f^{5} \cdot PS(f) \cdot df}.$$  (2)

SFI was calculated for each repetition, and the relative changes in values for each repetition were calculated against the first repetition of the corresponding set. The results of those muscles were averaged to obtain a single variable, which was then used in the statistical analyses. This process was performed using MATLAB R2020a (Mathworks, Natick, MA, USA).

## Velocity measurements

The velocity during exercise was recorded using a linear encoder (Fitro Dyne; FiTRONiC s.r.o., Bratislava, Slovakia). This encoder was placed directly under the dumbbell/weights and attached to weights through a cable. The sampling frequency was 100 Hz, and upward/downward displacement changes over time during the lifting were transferred to the computer and recorded with the assistance of proprietary software (Premium 3.0; FiTRONiC s.r.o., Bratislava, Slovakia). The mean concentric velocity was used in subsequent analyses. As different numbers of repetitions were required for each experimental condition of each participant, the first, median, and last repetition of experimental conditions were used for velocity analysis during three experimental conditions.

## Statistical analyses

The overall RPE and average SFI of the experimental conditions were analyzed using one-way analysis of variance (ANOVA). When analyzing RPE, SFI, and the mean velocity of the experimental trial, two-way (3 conditions × 3 timepoints) repeated ANOVA was used to test for main and interaction effects of conditions and time. Further, the Bonferroni post-hoc test was used to determine the significant effects and interactions between variables. As every participant performed a different number of repetitions, the first, median, and the last repetition from experimental conditions were used for 2-way ANOVA analysis.

Stepwise multiple regression analyses were performed with RPE, mean velocity, and SFI. The predictors were the RPE score obtained during exercise and the mean velocity, while the output variable was the SFI. The non-overlapping part of the three conditions was used in the analysis (*Migliaccio et al., 2018*). The determination coefficients obtained from regression analyses were interpreted as trivial (<0.02), small (0.02 to <0.13), medium (0.13 to <0.25), or large (>0.25) effects, according to *Cohen (1992)*. The variance inflation factors were used to assess multicollinearity. Overall, 163 EF and 235 KE trials were used in the multiple regression analysis. Statistical significance was acceptable at $p < 0.05$ in all analyses. The statistical analyses were performed using SPSS version 24.0 (SPSS Inc., Chicago, IL, USA).

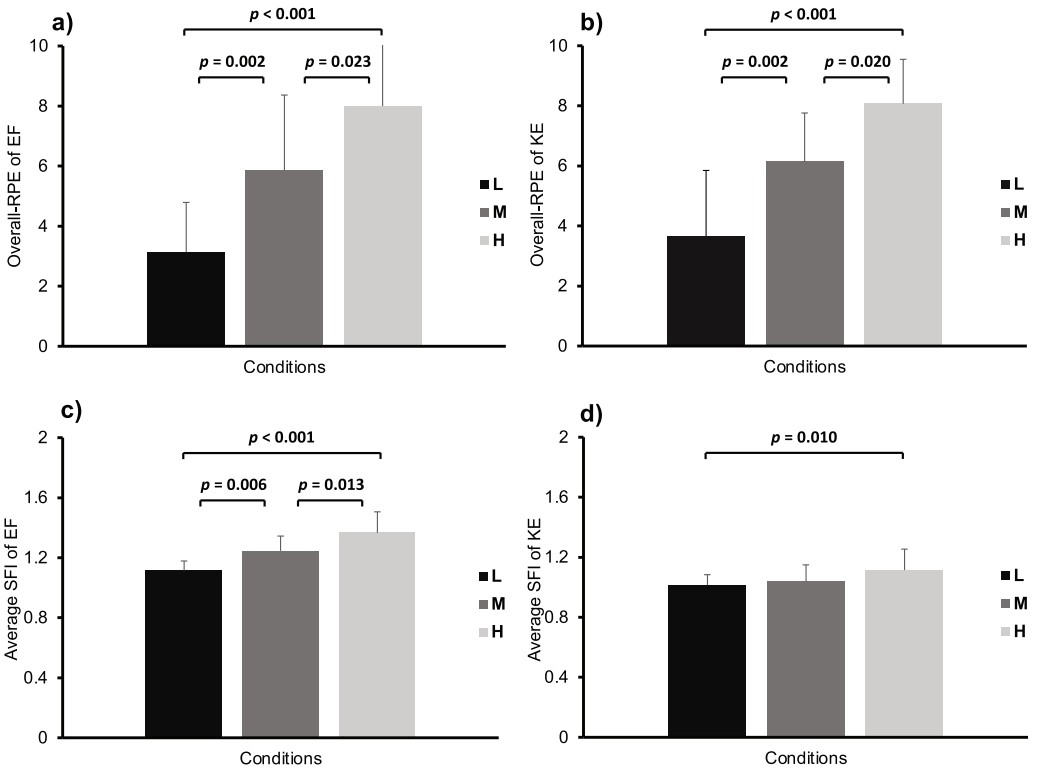

**Figure 1 Overall ratings of perceived exertion (RPE) and average spectral fatigue index (SFI) of elbow flexion (EF) and knee extension (KE) exercises.** The *p* value represents the differences and significance levels between experimental conditions.

## RESULTS

The descriptive data of participants (mean ± SD) were as follows: age (yr) 23.20 ± 3.00, body mass (kg) 72.96 ± 13.09, height (cm) 171.40 ± 5.84, fat (%) 16.92 ± 6.44. The 1RM strength of EF of the right side was 15.70 ± 4.46 kg, and the left side was 15.81 ± 4.23 kg; the 1RM of KE of the right side was 50.17 ± 14.13 kg, and the left side was 50.24 ± 13.54 kg. For the EF exercise, 12.60 ± 2.90 repetitions were performed for the anchoring trial, and 3.80 ± 1.08, 7.67 ± 1.68, and 11.40 ± 2.59 repetitions were determined for the L, M, and H conditions. For the KE exercise, 18.33 ± 10.05 repetitions were performed for the anchoring trial, and 5.60 ± 3.07, 10.73 ± 6.10, and 16.33 ± 9.17 repetitions were determined for the three experimental conditions.

Significant differences in overall RPE (EF: $F_{(2, 42)} = 20.491$, $p < 0.001$, $\eta^2 = 0.494$; KE: $F_{(2, 42)} = 21.279$, $p < 0.001$, $\eta^2 = 0.503$) were observed between the L (95% CI of EF [2.046–4.221]; 95% CI of KE [2.702–4.631]), M (95% CI of EF [4.779–6.954]; 95% CI of KE [5.169–7.098]), and H (95% CI of EF [6.912–9.088]; 95% CI of KE [7.102–9.031]) conditions (Figs. 1A and 1B). Similar results were observed in the average SFI for EF ($F_{(2, 42)} = 20.051$, $p < 0.001$, $\eta^2 = 0.488$, 95% CI of L [1.057–1.171]; 95% CI of M [1.189–1.303]; 95% CI of H [1.309–1.423]) (Fig. 1C). With regards to the SFI for KE, a significant increase ($F_{(2, 42)} = 5.228$, $p = 0.010$, $\eta^2 = 0.199$) was observed between H with L

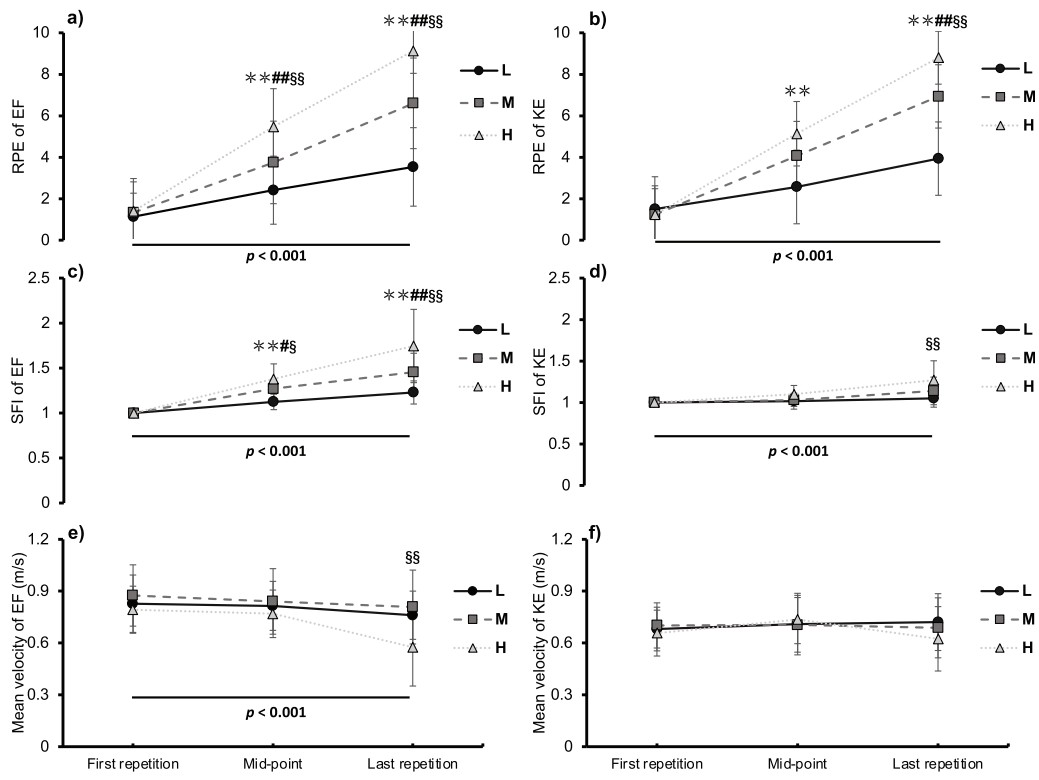

**Figure 2 Rating of perceived exertion (RPE), spectral fatigue index (SFI), and mean velocity during low (L, circle with solid lines), medium (M, square with dashed lines), and high (H, triangle with dotted lines) volume conditions of elbow flexion (EF) and knee ex.** An asterisk (*) indicates represents a significant difference in comparing L with other conditions, $p < 0.05$; two asterisks (**) indicate $p < 0.01$; # represents a significant difference in comparing M with other conditions, $p < 0.05$; ## $p < 0.01$; § represents a significant difference in comparing H with other conditions, $p < 0.05$; §§ $p < 0.01$; The $p$ value indicates the overall main effects and significant level for time.

(95% CI of L [0.969–1.061]; 95% CI of M [0.994–1.085]; 95% CI of H [1.069–1.160]) (Fig. 1D).

A significant overall main effect on time was observed in the RPE ($F = 182.862$, $p < 0.001$, $\eta^2 = 0.934$), SFI ($F = 53.536$, $p < 0.001$, $\eta^2 = 0.805$), and velocity ($F = 11.294$, $p < 0.001$, $\eta^2 = 0.465$) throughout the EF trial. Significant differences were observed in RPE and SFI between conditions at the mid-point (RPE: $F = 43.619$, $p < 0.001$, $\eta^2 = 0.770$, 95% CI of L [1.427–3.394]; 95% CI of M [2.800–5.093]; 95% CI of H [4.578–6.708]; SFI: $F = 24.007$, $p < 0.001$, $\eta^2 = 0.649$, 95% CI of L [1.073–1.177]; 95% CI of M [1.202–1.335]; 95% CI of H [1.280–1.486]) and the last repetition during EF exercise (RPE: $F = 74.911$, $p < 0.001$, $\eta^2 = 0.843$, 95% CI of L [2.448–4.618]; 95% CI of M [5.348–7.852]; 95% CI of H [8.510–9.757]; SFI: $F = 24.746$, $p < 0.001$, $\eta^2 = 0.639$, 95% CI of L [1.155–1.303]; 95% CI of M [1.334–1.575]; 95% CI of H [1.512–1.979]) (Figs. 2A and 2C). A significantly lower velocity ($F = 18.319$, $p < 0.001$, $\eta^2 = 0.567$, 95% CI of L [0.684–0.838] m/s; 95% CI of M [0.690–0.927] m/s; 95% CI of H [0.451–0.700] m/s) was observed in the H compared with the L and M conditions on the last repetition (Fig. 2E). With regard to KE, the overall main effect on time was observed in the RPE ($F = 437.174$, $p < 0.001$, $\eta^2 = 0.961$)

**Table 1  Output of multiple regression analyses.**

| Output-variable | Predictors | $R^2$ | Adj. $R^2$ | SEE | Constant (Sig.) | $\beta_{RPE}$ (Sig.) | $\beta_{Velocity}$ (Sig.) | F | Model sig. |
|---|---|---|---|---|---|---|---|---|---|
| SFI of EF | RPE, Velocity | 0.552 | 0.546 | 0.198 | 1.551 ($p < 0.001$) | 0.053 ($p < 0.001$) | −0.600 ($p < 0.001$) | 98.555 | $p < 0.001$ |
| SFI of KE | RPE, Velocity | 0.377 | 0.371 | 0.122 | 1.077 ($p < 0.001$) | 0.030 ($p < 0.001$) | −0.223 ($p < 0.001$) | 70.101 | $p < 0.001$ |

Note:
SFI, spectral fatigue index; EF, elbow flexion; KE, knee extension; RPE, ratings of perceived exertion; SEE, standard error estimators; $\beta$, coefficients.

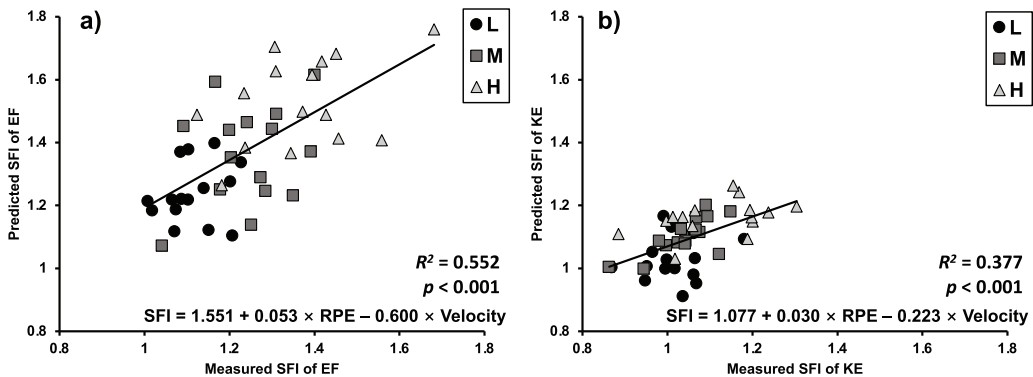

**Figure 3  Measured *vs* predicted spectral fatigue index (SFI) of elbow flexion (EF) and knee extension (KE) exercises.** The $p$ value represents the significant level of linear regression models.

and SFI ($F = 16.044$, $p < 0.001$, $\eta^2 = 0.534$). There were significant differences in the RPE between the experimental conditions at the mid-point ($F = 20.977$, $p < 0.001$, $\eta^2 = 0.600$, 95% CI of L [1.549–3.585]; 95% CI of M [3.114–5.019]; 95% CI of H [4.242–6.024]) and the last repetition ($F = 86.838$, $p < 0.001$, $\eta^2 = 0.861$, 95% CI of L [2.919–4.947]; 95% CI of M [6.059–7.808]; 95% CI of H [8.069–9.531]) (Fig. 2B). A significantly higher SFI ($F = 8.184$, $p = 0.006$, $\eta^2 = 0.369$, 95% CI of L [0.989–1.111]; 95% CI of M [1.044–1.235]; 95% CI of H [1.134–1.403]) was observed in the H compared with the L and M conditions for the last repetition (Fig. 2D). Significant interactions (conditions × times: $F = 3.695$, $p = 0.009$, $\eta^2 = 0.209$) were observed in exercise velocity during KE; however, pairwise comparisons did not indicate any difference between time and conditions (Fig. 2F).

The results of the regression analysis are shown in Table 1. The models were significant ($p < 0.001$), and determination coefficients were large in both EF ($R^2 = 0.552$) and KE ($R^2 = 0.377$). The scatter plots of the predicted SFI and the measured SFI are shown in Fig. 3.

## DISCUSSION

The purpose of this study was to investigate the validity of using RPE for assessing exercise-induced muscle fatigue and to establish prediction models for fatigue in resistance exercise situations. The most important contributions of this study are that it: (1) confirmed RPE as an effective tool for assessing exercise-induced muscle fatigue, (2) established two prediction models for assessing muscle fatigue by using the RPE score

and velocity during single-joint resistance exercises, and (3) suggested that velocity changes may not reflect muscle fatigue correctly when exercise target is no longer explosive performance.

During dynamic exercises, effort sensation increases in response to both central and peripheral changes, such as motor unit recruitment, firing rate, and anaerobic glycolysis (*Lagally et al., 2002b*; *Amann et al., 2011*; *de Morree, Klein & Marcora, 2012*; *Blain et al., 2016*). For instance, *Lagally et al. (2002b)* revealed that sEMG and blood lactic levels changed correspondingly, with results similar to RPE under different intensities. In our study, the relative intensity was designed to be constant between conditions; thus, the possibility that RPE was dependent on intensity could be eliminated. The cause of the difference in RPE can be attributed to the difference in duration of consecutive exercises. Previous research indicated that more repetitions could have an impact on glycolytic metabolism and RPE, even when the total rest time remain constant (*Esteban et al., 2012*; *Mayo, Iglesias-Soler & Kingsley, 2019*; *Zhao, Yamaguchi & Okada, 2020*). Consistent with these previous findings, we observed that a greater number of repetitions were performed continuously with increasing volume, which might have induced more severe peripheral perturbation, such as hydrogen ion concentration in skeletal muscles. These contraction-induced mechanical and chemical changes can stimulate afferent feedback from molecular receptors on the terminal end of myelinated (group III) and unmyelinated (group IV) nerve fibers in skeletal muscle, increasing the perception of exertion (*Amann et al., 2011*; *Blain et al., 2016*; *Broxterman et al., 2018*). As a result, RPE was higher during this trial which was statistically significant. Moreover, the dramatic disruption in homeostasis interfered with the conduction velocity of the muscle fiber (*Brody et al., 1991*), indicated by the shifting of the spectral power frequency to a low frequency band and a significant increase in the SFI. These corresponding changes suggest that the RPE reflects fatigue responses of single-joint resistance exercises.

When the results of the upper and lower body were compared, fatigue in the lower body was more difficult to predict. The upper body exhibited a higher level of $R^2$ than the lower body (EF: $R^2 = 0.552$; KE: $R^2 = 0.377$). This may be attributed to the differences in muscle fiber composition. Some studies have indicated that there are larger type-I fiber distributions in the vastus lateralis compared to the biceps and triceps brachii (*Miller et al., 1993*; *Mygind, 1995*). Moreover, these differences in fiber distribution could lead to variations in lactate kinetics in the arms and legs (*Van Hall et al., 2003*). The values of the SFI during the H condition were consistent with this inference, where arm muscles displayed unignorable fatigue faster than the legs (mid-point *vs* last repetition of the H condition). Thus, the different fatigue development patterns in the arms and legs may be due to muscle fiber distribution differences, which might have changed the accuracy of the model.

As for the results of velocity, a significant difference was only observed at the end point of EF exercise, whereas no significant difference was observed between conditions at any time point during KE exercise. It seemed that the experimental designs with non-explosive, non-failure, and single-joint settings were unsuitable for using velocity changes

as measures of fatigue. In previous research that focused on the relationship between velocity and fatigue-related measures, multi-joint exercises, explosive manner, and until-failure design were always selected and performed (*Sánchez-Medina & González-Badillo, 2011*; *Evandro et al., 2019*; *Mayo, Iglesias-Soler & Kingsley, 2019*). To assess the validity of velocity changes as a muscle fatigue indicator in various resistance exercise situations, we used single-joint exercises, a non-explosive manner, and a non-failure design. In these situations, the validity of velocity changes seemed to have very limited precision for mechanical measurement of fatigue. Although the lack of familiarity with unilateral exercises and the deviations in weight determination (nearest kg for EF exercise and nearest 5 lbs. for KE exercise) may have had some influence on velocity measures, the volume of the H condition observed (90% of until-failure volume) was considered to be very close to the volitional failure, and we expected to observe velocity changes corresponding to the experimental conditions. Although significant interaction (EF: $p < 0.001$; KE: $p = 0.009$) was observed, similar significant changes in velocity corresponding to SFI failed to occur during *post hoc* test. Similar changes in velocity are likely to be observed if a larger sample size was recruited. Nevertheless, we can conclude that velocity changes are inappropriate for fatigue assessment of non-explosive, non-failure, and single-joint resistance exercises.

When using velocity changes as a muscle fatigue indicator in sports exercise and clinical situations, only explosive, until-failure, and multi-joint settings are acceptable. Some training programs should be avoided, such as open kinetic chain and muscular hypertrophy-aimed exercises. For instance, it is possible that muscle fatigue appeared while velocity changes failed to occur. For lower-body exercises, more caution should be observed when using only velocity changes as an assessing tool in the fatigue prediction process. This is consistent with the rationale offered earlier concerning muscle fiber distribution. It is desirable to assess muscle fatigue by combining the RPE score with velocity changes during resistance exercise situations. If Borg's CR-10 scale is the only available tool in the fatigue assessment process, muscle fatigue in the arm would appear around an exertion level ranging from "Moderate" to "Somewhat strong". For the leg, significant muscle fatigue would appear around the "Very strong" level.

The current study has several limitations. First, the metabolic and endocrine variables were not assessed in this study. Some studies have noted that training volume might translate into different neuroendocrine system perturbations and/or neuromuscular function changes, potentially affecting perceived exertion (*Hiscock et al., 2018*). Second, the present study only assessed single-joint exercises. Different results might be observed when performing multi-joint exercises because more muscle groups will be recruited. Further, the findings may not offer insights on contribution of gender as a factor, as all participants were male. *Otto, Emery & Côté (2018)* revealed that sex differences affect fatigue adaption strategies in certain muscles. Accordingly, future studies should focus on the metabolic and endocrine measures, multi-joint exercises, and sex-specific effects of RPE and muscle fatigue.

## CONCLUSIONS

The current study demonstrated that the SFI and RPE changed correspondingly, which revealed a link between perceptual responses and muscle fatigue. We concluded that muscle fatigue exhibits similar increases in perceptual responses when resistance exercise is performed. These results demonstrate that RPE is a useful tool for assessing fatigue during single-joint resistance exercises. Furthermore, we offer two significant models for predicting muscle fatigue by using RPE score and velocity in clinical and sports situations. However, velocity changes have only limited precision on fatigue assessment when the exercise target is no longer explosive performance, and more caution should be used when predicting muscle fatigue in the legs. It is recommended to combine RPE responses with velocity changes and assess muscle fatigue comprehensively during clinical and sports situations.

### Funding

This study was supported by JSPS KAKENHI Grant Number JP (19K11452). The first author is subsidized by China Scholarship Council under Grant Number (201908050185). The funders had no role in study design, data collection and analysis, decision to publish, or preparation of the manuscript.

### Grant Disclosures

The following grant information was disclosed by the authors:
JSPS KAKENHI: 19K11452.
China Scholarship Council: 201908050185.

### Competing Interests

The authors declare that they have no competing interests.

### Author Contributions

- Hanye Zhao conceived and designed the experiments, performed the experiments, analyzed the data, prepared figures and/or tables, authored or reviewed drafts of the paper, and approved the final draft.
- Takuya Nishioka conceived and designed the experiments, authored or reviewed drafts of the paper, and approved the final draft.
- Junichi Okada conceived and designed the experiments, authored or reviewed drafts of the paper, and approved the final draft.

### Human Ethics

The following information was supplied relating to ethical approvals (*i.e.*, approving body and any reference numbers):

The study was approved by the Waseda University Human Ethics Committee (No. 2020-042).

## Data Availability

The processing code of surface electromyography are available in the Supplemental Files.

## Supplemental Information

Supplemental information for this article can be found online at http://dx.doi.org/10.7717/peerj.13019#supplemental-information.

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
