# Peer review of "Validity of using perceived exertion to assess muscle fatigue during resistance exercises"

_PeerJ, doi:10.7717/peerj.13019_

## Round 0.1 · original submission · Minor Revisions

Both reviewers found merit in your manuscript but both made suggestions for improvement.

Reviewer 1 ·

Basic reporting

No comments. See PDF

Experimental design

No comments. See PDF

Validity of the findings

No comments. See PDF

Additional comments

No comments. See PDF

Annotated reviews are not available for download in order to protect the identity of reviewers who chose to remain anonymous.

Reviewer 2 ·

Basic reporting

no comment

Experimental design

1. I believe the paper could benefit from including more data on volume. Volume was determined as number of repetitions performed at 65% 1RM, and number of repetitions was determined as 30, 60 and 90% of maximal repetitions from an initial maximal repetition test. Is that correct? It might be interesting to report the mean, SD and range for repetitions at each level for each exercise, and for the maximal repetition test.
2. Do you think performing the maximal repetition test immediately prior to experimental trials might have affected exertion? Could the maximal repetition test have been performed in the orientation session? Why did you choose to do anchoring procedures with the maximal repetition test rather than with the 1RM test?
3. Lines 187-190 indicate that some participants had difficulty maintaining the pace. The authors explanation of what occurred in this situation could use some clarification. Weren't participants already pausing between repetitions? Were participants only struggling to maintain pace during the concentric portion?
4. The authors use the terms "velocity loss" and "velocity changes" throughout the paper. As velocity is included in the prediction models, it might be helpful to more clearly differentiate between the two terms, and to clarify the velocity that is used in the models vs. "velocity loss".

Validity of the findings

no comment

Additional comments

This paper contributes to the literature in this area and I appreciate the goal of facilitating determination of fatigue for practitioners. The results could be very applicable in practice.

1. In the abstract, line 30, suggest changing the word "exhibited". Resulted in?
2. I found lines 72-76 a bit confusing; perhaps reword? I understand the idea of velocity loss validity during only explosive exercise, but then it is unclear how the next statement is an example? Are you saying that for hypertrophy, explosive exercises are less ideal and so velocity loss is not a good indicator of fatigue in that setting? Perhaps the issue is just with the term "for example". Maybe "However"? Or I may be misunderstanding...
3. Line 86 - what is meant by the "shape" of the RPE scale? I was curious on your choice of the CR-10 scale... so perhaps this is a place to explain that choice? There are also a variety of task-specific OMNI scales, although it seems those weren't used here, which is fine, but could use an explanation.
4. Do the authors feel that RPE alone reflects fatigue? Or only in conjunction with velocity? Line 366 might suggest that RPE alone reflects fatigue - perhaps some number above 8 on the CR-10 scale? That is the general suggestion for aerobic activities. Could the authors comment on the use of RPE alone?
5. In lines 379-394, I might suggest different words for "unignorable" "un-failure". I'll admit I don't have great suggestions and ultimately it makes sense. Perhaps reword lines 381-2 ... I think the authors are saying that the experimental design of non-explosive, single joint activity was not suitable for using velocity loss as a measure of fatigue... and therefore, exercise in that type of setting would not benefit from using velocity loss. Perhaps un-failure could be reworded as "not to failure" design. Not sure that's any better :).

---

## Round 0.2 · accepted · Accept

The reviewers found your changes satisfactory.

Reviewer 1 ·

Basic reporting

All of my comments and concerns have been addressed. The newest edition of the paper is sufficient for publication.

Experimental design

All of my comments and concerns have been addressed. The newest edition of the paper is sufficient for publication.

Validity of the findings

All of my comments and concerns have been addressed. The newest edition of the paper is sufficient for publication.

Additional comments

All of my comments and concerns have been addressed. The newest edition of the paper is sufficient for publication.

Reviewer 2 ·

Basic reporting

No Comment

Experimental design

No comment

Validity of the findings

No comment

Additional comments

The changes made by the authors in response to reviewers strengthens the paper and addresses major concerns.